# A Critical Comparison of Correlations for Rapid Estimation of Subgrade Stiffness in Pavement Design and Construction

**Christina Plati \*** and **Maria Tsakoumaki**

Laboratory of Pavement Engineering, School of Civil Engineering, National Technical University of Athens, GR-15773 Athens, Greece
* Correspondence: cplati@central.ntua.gr

**Abstract:** According to pavement design principles, the subgrade or soil layer serving as the foundation for pavement depends on the properties and stiffness of the soil material. The resilient modulus ($M_R$) is the absolute measure of the bearing capacity of the subgrade for pavement design. However, due to the complexity of $M_R$ testing, indirect methods are chosen to determine $M_R$. In this context, the CBR test is considered a practical tool for determining the strength of the subgrade, but the use of the correlations of $M_R$-CBR has caused great controversy in the scientific community. Nevertheless, such correlations are widely used in practice for pavement design, and the question of their influence on design results is always raised. Therefore, the present study investigates the use and applicability of the $M_R$-CBR correlations for the subgrade with respect to the design of flexible pavements, with the aim of optimizing the thickness and bearing capacity of the unbound base/sub-base. Based on the previous debate on the use of $M_R$-CBR correlations, this study first investigates the main correlations between $M_R$ and the CBR index based on a thorough review of the literature. Using the properties of certain medium-grained soils used in practice and the corresponding values of $M_R$, estimated by the various $M_R$-CBR correlations reported in the international literature, a theoretical pavement design is then carried out on the basis of a sensitivity analysis. A major outcome of the sensitivity analysis is the identification of the most optimal correlation for estimating $M_R$ in pavement design, while the development of a global $M_R$-CBR correlation applicable to most types of soil materials used in pavement construction remains an important topic for future research.

**Keywords:** soil materials; subgrade layer; bearing capacity; CBR; resilient modulus; flexible pavement design

## 1. Introduction

According to pavement design principles, the subgrade layer functions as the foundation for the whole structure of the pavement section [1,2]. Therefore, the performance of the pavement is based on the strength and bearing capacity of the subgrade layer. There are numerous tests that have been used for the determination of the subgrade's quality. The most accurate and precise of these tests is considered to be repeated triaxial testing, which expresses the soil bearing capacity through the resilient modulus $M_R$. It was introduced by Seed et al. [3] and is still used in pavement design, as required by numerous official guides [1,4–6]. However, because the use of direct methods, i.e., laboratory tests, to measure $M_R$ is complex, costly, and time-consuming, many studies have focused on finding alternative indirect methods to calculate or approximate $M_R$ values. The most commonly used models to indirectly determine $M_R$ often depend on empirical correlations, particularly between $M_R$ and the California bearing ratio (CBR), which indicates the resistance of a compacted material to the penetration of a cylindrical load surface [7].

These correlations have generated much debate in the scientific community over time [8,9]. In particular, even from the very first years of pavement engineering, Hight and Stevens [10] and Fleming and Rogers [11] pointed out that the CBR value is a strength

value rather than a support value in relation to the recovery behavior of materials, while Brown et al. [12] demonstrated that resilient modulus is not a simple function of the CBR index, but depends on the soil type and stress state. Hossain et al. [13] observed the differences in the methods used by different agencies, such as the size of the mold, compaction techniques, and effort, and it was found that the correlations between the values of resilient modulus and all test results, including CBR, were not statistically significant. Kumar et al. [14] also found that the two parameters were different in nature. His study states that $M_R$ is estimated by dynamic tests and depends on the stress state, while CBR is based on point load tests. Mendoza and Caicedo [15] show that in the CBR test, the stresses and strains within the material, including the areas that behave in the elastic and plastic range, are not homogeneous. This lack of homogeneity limits the correct mechanical interpretation of results following a $M_R$-CBR correlation. On the other hand, Garg et al. [16] considered the possibility that CBR value can be converted into the resilient modulus with considerable dispersion. Later, Dione et al. [17] noticed that the correlations between $M_R$ and CBR could be used for estimating the $M_R$ value, but they should be used carefully, as they tend to ''over-predict'' or ''under predict'' the $M_R$. Finally, most recent studies [18–21] highlight that the prediction of $M_R$ should be more accurate if it is correlated with the soil properties.

Overall, it is generally accepted that the subgrade or soil layer used as the foundation for pavement depends on the properties and stiffness of the soil material [22]. $M_R$ is the absolute measure of subgrade bearing capacity for pavement design. However, due to the $M_R$ tests complexity, indirect methods are chosen for $M_R$ determination. In this context, the CBR test is considered to be a practical tool for determining the strength of the subgrade, but use of the correlation of $M_R$-CBR has caused great controversy in the scientific community. Nevertheless, such correlations are widely used in practice for pavement design, and the question of their influence on design results is always raised. Therefore, this study investigates the use and applicability of $M_R$-CBR correlations for the subgrade in terms of flexible pavement design to optimize the thickness and bearing capacity of the unbound base/sub-base. Based on the previous debate on the use of $M_R$-CBR correlations, this study first investigates the main relationships between $M_R$ and the CBR index through a thorough literature review. Using the properties of certain medium-grained soils used in practice and the corresponding values of $M_R$, estimated by the various $M_R$-CBR correlations reported in the international literature, a theoretical pavement design is then carried out on the basis of a sensitivity analysis. Taking into account the limitations of each correlation, we evaluate which correlations are applicable to the subgrade of pavements (subsoil) regardless of soil gradation.

## 2. Methodology

### 2.1. General Description

The current research methodology includes four parts: (i) the literature review on $M_R$-CBR correlations, (ii) the laboratory procedure, (iii) the definition of pavement design parameters and (iv) the sensitivity analysis. Firstly, a thorough research was conducted in order to note the most common $M_R$-CBR correlations in the global literature. Then, during the laboratory procedure, the physical properties of the studied soil were determined by grain size analysis and a compaction test, while the mechanical properties were determined by CBR tests. Based on the CBR results, the $M_R$ value was determined for each $M_R$-CBR correlation found in the international literature. These estimated values of $M_R$ were used in the next part of the current methodology, where a theoretical pavement design was performed based on certain design parameters, described in Section 2.4. Based on the corresponding results, a sensitivity analysis was performed with respect to the thickness and bearing capacity of the base and sub-base, which were considered as a single layer. In Figure 1, the methodology followed in the current research is presented.

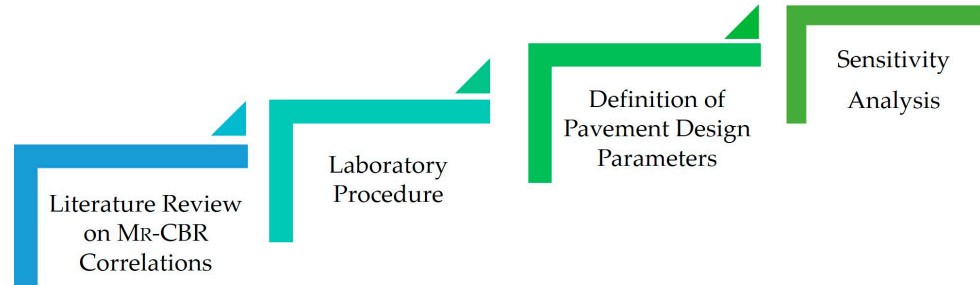

**Figure 1.** Research methodology.

In the next sections, each step of the research methodology is presented more thoroughly, while the sensitivity analysis, including the discussion of the results and the conclusions, follows.

### 2.2. Literature Review on $M_R$-CBR Correlations

Beyond the doubts concerning the correlations between $M_R$ and CBR, they are still widely used in practice for pavement design, as the CBR test is one of the simplest tests and it is still considered to be the most common indentation test to check the quality of subgrade and granular material [23]. In this context, it should be noted that the field bearing capacity of the subgrade of road pavements and thus the CBR value in situ should be better with proper compaction, otherwise poor results are observed [24,25]. Thus, the California bearing ratio (CBR) test was developed in 1929 by the Department of Roads in California to evaluate the behavior of pavements by testing a large number of aggregates. Later, this method was also applied to soil materials [26]. In laboratory CBR tests, soil samples are compacted in a cylindrical mold and then loaded with a stamp on the soil surface. The results are then used to calculate the CBR value as the percentage ratio between the load (P) measured at specific displacements and the standard loads ($P_T$) at those displacements, as shown in Equation (1).

$$\mathrm{CBR} = \frac{\mathrm{P}}{\mathrm{P_T}} \times 100\% \tag{1}$$

As for the resilient modulus $M_R$, it was first proposed in the 1950s in California [7]. Hveem [27] related the elastic properties of foundation layers to road surface failure due to surface cracking, while Seed et al. [3] introduced the definition of the parameter, which is still valid today, performing repeated load triaxial tests on compacted soil samples. During each test, soil specimens are placed in a cylindrical chamber, on which a lateral stress ($\sigma_3$) is imposed. An axial stress ($\sigma_1$) is then imposed in a tri-axial chamber, at a repetitive rate, with intermediate pauses and unloading of the test specimen, in order to simulate traffic load and the passage of vehicles. Eventually, after each test, the residual deformation ($\varepsilon_\alpha$) of specimen is measured and $M_R$ is determined as shown in Equation (2).

$$M_R = \frac{\sigma_1 - \sigma_3}{\varepsilon_\alpha} \tag{2}$$

Historically, there have been many developed relationships that correlate the two parameters of bearing capacity [28]. These relationships are used to calculate $M_R$ through the CBR index. Heukelom and Foster [29] developed one of the first correlations between $M_R$ and CBR. Dynamic tests were conducted using samples from different soil types of which subgrade layers generally are composed. Additionally, it should be noted that they adopted an elastic theory, but soils are not purely elastic materials and deformations due to CBR testing are mostly plastic with very little elastic rebound. Finally, the test results led to Equation (3), which is applicable for CBR values varying from 2 to 200%.

$$M_R(\mathrm{psi}) = 1565 \times \mathrm{CBR} \quad M_R(\mathrm{MPa}) = 10.8 \times \mathrm{CBR} \tag{3}$$

Later, Heukelom and Klomp [30] performed Rayleigh wave propagation tests and dynamic impedance tests in Netherlands and the United Kingdom to determine the modulus of elasticity, while dynamic compaction tests were conducted in order to determine the CBR value. Based on test results, it emerged that modulus of elasticity was 750 to 3000 times the CBR value and, eventually, Equation (4) was proposed. This equation provides reasonable values of $M_R$ for fine-grained soils with a soaked CBR value of 10 or less.

$$M_R(\text{psi}) = 1500 \times \text{CBR} \quad M_R\,(\text{MPa}) = 10 \times \text{CBR} \tag{4}$$

Green and Hall [31] proposed a different correlation between the two parameters, as shown in Equation (5), based on results that came out after similar dynamic tests on experimental roads. As for flexible pavements, where the subgrade layer is formed from clay, namely fine-grained soils, according to test results, the CBR values of the soil materials are low, with maximum values up to 7%.

$$M_R(\text{psi}) = 5409 \times \text{CBR}^{0.71} \quad M_R(\text{MPa}) = 37.3 \times \text{CBR}^{0.71} \tag{5}$$

The next correlation was assumed to be developed under the auspices of the South African Council on Scientific and Industrial Research [32]. The authors used equations of the form $M_R = k \times \text{CBR}$ by modifying the k factor, which depends on soil type and laboratory tests regarding soil quality. Eventually, Equation (6) was proposed and is applicable for CBR values ≤20%.

$$M_R(\text{psi}) = 3000 \times \text{CBR}^{0.65} \quad M_R(\text{MPa}) = 20.7 \times \text{CBR}^{0.65} \tag{6}$$

Powel et al. [33] proposed Equation (7). It was based on in situ CBR tests and Ray-leigh wave propagation tests, and it is applicable for CBR values varying from 1 to 12%.

$$M_R(\text{psi}) = 2555 \times \text{CBR}^{0.64} \quad M_R(\text{MPa}) = 17.6 \times \text{CBR}^{0.64} \tag{7}$$

Hopkins et al. [34] developed a new $M_R$–CBR correlation, analyzing again the data used by Heukelom and Klomp [30]. Thus, Equation (8) was proposed and was applied to fine-grained soils with very low CBR values ≤ 10%, as laboratory tests were performed on clay and clay-sand soils.

$$M_R(\text{psi}) = 2596 \times \text{CBR}^{0.874} \quad M_R(\text{MPa}) = 17.9 \times \text{CBR}^{0.874} \tag{8}$$

In summary, all researchers have used laboratory data to develop correlations that correlate CBR with the resilient modulus. However, each correlation is applicable to a specific range of CBR values. The correlation of Heukelom and Klomp [30] and Powell et al. [33] applies to CBR values of less than 10%, while the correlation of Heukelom and Foster [29] applies to a wide range of CBR values from 2 to 200%. It is worth noting that almost all correlations are limited to fine-grained soil material. Therefore, all correlations are applicable to a specific soil grain size. However, there is a wide range of soils of different grain sizes that vary from place to place where a pavement is to be built. In this case, it is doubtful whether these correlations can be applied to materials of different grain sizes.

### 2.3. Laboratory Procedure

During the experimental process, sieve analysis, compaction and CBR tests were conducted in the present study. It is worth mentioning that the studied soil material did not belong to the soil categories used to develop the $M_R$-CBR correlations from the literature review. Therefore, sieve analysis was carried out according to EN 933-2 [35] to determine the soil gradation. In the laboratory, the used sieves were sized in descending order of 63 mm, 40 mm, 31.5 mm, 16 mm, 8 mm, 4 mm, 2 mm, 1 mm, 0.5 mm and 0.063 mm.

Additionally, the coefficient of uniformity $C_u$ and the coefficient of curvature $C_c$ were estimated through Equations (9) and (10), respectively.

$$C_u = \frac{d_{60}}{d_{10}} \tag{9}$$

$$C_c = \frac{d_{30}^2}{d_{60}d_{10}} \tag{10}$$

where $d_{60}$ is the grain diameter at which 60% of mixture particles are finer, $d_{10}$ is the grain diameter at which 10% of mixture particles are finer, and $d_{30}$ is the grain diameter at which 30% of mixture particles are finer.

These coefficients are indicative of the shape of the gradation curves and they show if a material is well-graded. According to ASTM D2487-17E01 [36], a material is considered to be well graded if $C_u$ is greater than 4 for coarse material or 6 for finer material, including sand, and $C_c$ is between 1 and 3.

Afterwards, a compaction test was performed to determine the optimum moisture content (OMC) as the moisture content corresponding to the maximum dry density (MDD). This test was based on the modified Proctor test of European specification EN 13286-2 [37]. For this test, around 5 kg of the examined soil was used. The material was placed in the mold in three layers. For each layer, a mass of about 1.6 to 1.8 kg was used, while 28 blows were applied with the rammer of 4.5 kg to compact each layer and then determine the dry density. This procedure was repeated for different values of moisture content.

The CBR tests were performed in accordance with EN 13286-47 [38]. Once the OMC was determined, six specimens of the tested soil were again compacted at the OMC so that the CBR test could be performed. Each specimen was placed on the loading device. Figure 2 shows the CBR testing equipment.

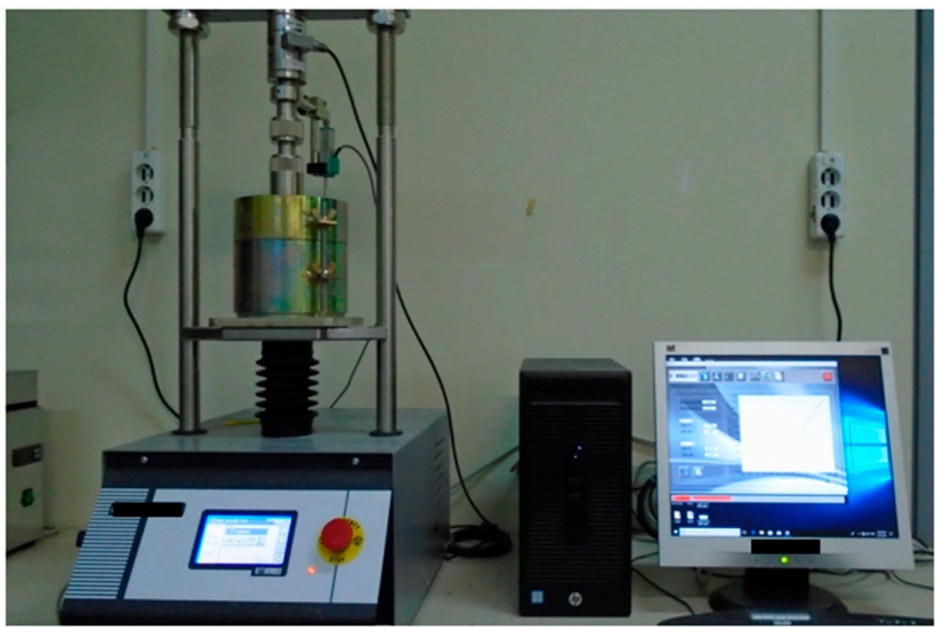

**Figure 2.** Equipment for CBR testing.

During the test, the load was applied to the penetration piston so that the penetration rate was constant at about 1.27 mm/min. The load value was recorded for each penetration into the material. The force–penetration curve was recorded for each test. The CBR value was determined based on Equation (1), where P is the applied load that causes penetration to the sample equal to 2.5 mm and 5 mm, respectively, and $P_T$ is the standard load (13.2 and 20 kN) corresponding to penetrations of 2.5 and 5 mm, respectively.

## 2.4. Definition of Pavement Design Parameters

In this section, the design parameters of the theoretical pavement sections are detected and defined. Particularly, the current study focuses on the relation between the unbound and the subgrade layer in the design of flexible pavements. It is well known that, according to the principles of flexible pavement design, the thicknesses of asphalt layers and the base/sub-base course are determined based on the bearing capacity of the subgrade. For estimating the subgrade bearing capacity, it has already been stated that the most reliable parameter is the resilient modulus $M_R$, which is a measure of subgrade material stiffness. For the purpose of this study, $M_R$ was estimated using the $M_R$-CBR correlations found in the international literature. In addition, we considered the theoretical pavement sections to be high-volume roads (HVR), namely highways, while the traffic loads were considered to be the same in all discussed cases.

In addition, considering that the cross-section of a pavement is multi-layered, which makes the design process complex, it must be simulated in a simpler form. There are three major static simulation models [39]: two layers, three layers, and four layers, as shown in Figure 3, which specifies the thickness ($h_i$) and elastic modulus ($E_i$) of each layer.

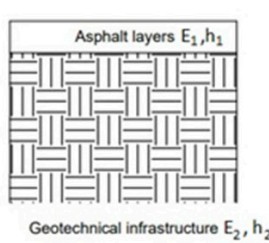
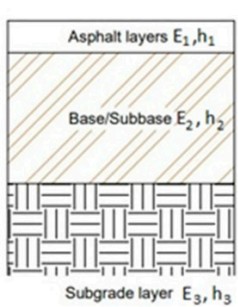
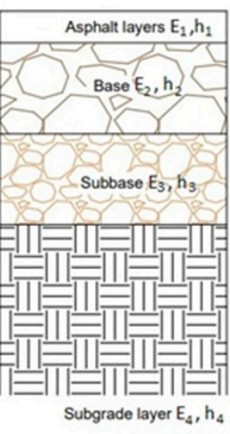

Static simulation model of 2 layers    Static simulation model of 3 layers    Static simulation model of 4 layers

**Figure 3.** Static simulation models for flexible pavement structure.

In this study, a 3-layer system was used for pavement design. In the design of flexible pavements, international experience [40] relates the modulus of elasticity of layers of the unbound material to the modulus of elasticity of the underlying layer, namely the soil layer. In the case of a simulated 3-layer structure, this correlation is thus expressed by Equation (11).

$$E_2 = k \times E_3 \text{ (MPa)} \tag{11}$$

where $E_2$ is the modulus of elasticity of the base/sub-base and $E_3$ is the modulus of elasticity of the subgrade, namely $E_3 = M_R$, and k is a coefficient that indirectly expresses the degree of compaction with respect to the bearing capacity of the layers involved and is estimated using Equation (12).

$$k = 0.2 \times h_2^{0.45} \text{ (mm)} \tag{12}$$

where $h_2$ is the thickness of the layer of the base/sub-base in mm.

As for the coefficient k, it has a limited range of values and is indicative of the possible failure of the pavement. In general, an excessive increase in k or a large deviation between $E_2$ and $E_3$ leads to the development of tensile stresses at the surface between the layer of base/sub-base and the subgrade. The explanation for this is that a high-strength layer, which is the top layer of the subgrade, cannot be properly compacted on a soft soil layer. In addition, the coefficient k should not be very low, as this may lead to early failure due to continuous compaction of the unbound layer at the beginning of the service life. As

for the pavement performance, there is a controversial situation between the two main failure mechanisms: fatigue cracking and total rutting. As Nassiri et al. [41] found, fatigue cracking is more sensitive to the base/sub-base modulus ($E_2$) and its increase causes a reduction in fatigue cracking and a subsequent increase in the ratio between $E_2$ and $E_3$. On the other hand, total rutting is more sensitive to the subgrade modulus ($E_3$) and its increase causes the reduction of total rutting as well as reduction in the ratio between $E_2$ and $E_3$. Hence, it is very critical to balance these two cases in order to choose the adequate parameters/design inputs (k, $E_2$, $E_3$) for proper pavement performance. In Table 1, some correlations for the estimation of k are presented.

**Table 1.** Values of $\frac{E_2}{E_3} = k$ according to global literature.

| Reference | Equation | |
|---|---|---|
| Heukelom and Foster [29] | $\frac{E_2}{E_3} = 2.5$ | (13) |
| Dormon and Metcalf [42] | $2 < \frac{E_2}{E_3} < 4$ | (14) |
| Brown and Pappin [43] | $1.5 < \frac{E_2}{E_3} < 7.5$ | (15) |

More specifically, Heukelom and Foster [29] observed that horizontal tensile stresses developed in the lower part of the unbound material layer when the ratio between the elastic modulus $E_2$ of the unbound material layer and the elastic modulus $E_3$ of the soil layer (subgrade) was more than 2.5. Due to the passage of vehicles, these stresses may lead to softening of this layer. This practically means that $E_2$ is reduced to a limit where no tensile stresses occur. In addition, Brown and Pappin [43] relied on more rigorous nonlinear finite element analyses and they noticed that the moduli ratio increased with increasing thickness $h_2$, while the subgrade modulus remained constant. Moreover, according to Dormon and Metcalf [42], the relationship between the moduli ratio (k) and the equivalent thickness of the base/sub-base layer ($h_2$) is linear and the values are analogous. In this case, when the resilient modulus of the subgrade had a constant value, as is the case for any correlation $M_R$-CBR, the modulus of the unbound granular layer increased with increasing thickness. This finding was later confirmed by Uzan [44]. In his conclusions, Uzan [44] demonstrates that $E_2$ increases as the thickness of the granular layer increases.

In this context, the Australian Road Design Guide [6] recommends that the thickness of the layer of unbound material should be between 50 and 150 mm, excluding the coefficient k. Following this recommendation, Moffat and Jameson [45] suggested that the values for the modulus of elasticity of the layer of unbound material should be 500 and 350 MPa for high and normal traffic loads, respectively. In Figure 4, the results of the correlations suggested by Dorman and Metcalf [42], Brown and Pappin [43] and Moffat and Jameson [45] are compared.

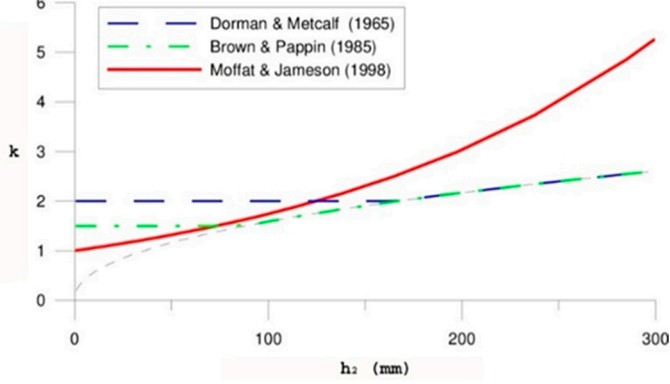

**Figure 4.** Comparison of k from alternative correlations [46], namely Dorman and Metcalf [42], Brown and Pappin [43] and Moffat and Jameson [45].

As Figure 4 shows, the respective k values resulting from the three correlations are similar in a range of overlying layer thickness $h_2$ from 75 to 150 mm. In this case, it can be deduced that the stiffness approximately doubles with every 125 mm of granules applied.

All of the above correlations were considered in the sensitivity analysis presented in the following section.

## 3. Results and Discussion

### 3.1. Laboratory Procedure

3.1.1. Physical Properties

Following the grain size analysis and the compaction test for the tested soil, its physical properties are presented in Table 2 and the grain size distribution is depicted in Figure 5.

**Table 2.** Physical properties of the tested soil.

| Properties | Sample Soil |
|---|---|
| $d_{10}$ (mm) | 0.2 |
| $d_{30}$ (mm) | 2.2 |
| $d_{60}$ (mm) | 10 |
| Coefficient of uniformity ($C_u$) | 4.5 |
| Coefficient of curvature ($C_c$) | 2.4 |
| Optimum moisture content (%) | 6.5 |
| Maximum dry density (kg/m$^3$) | 2240 |

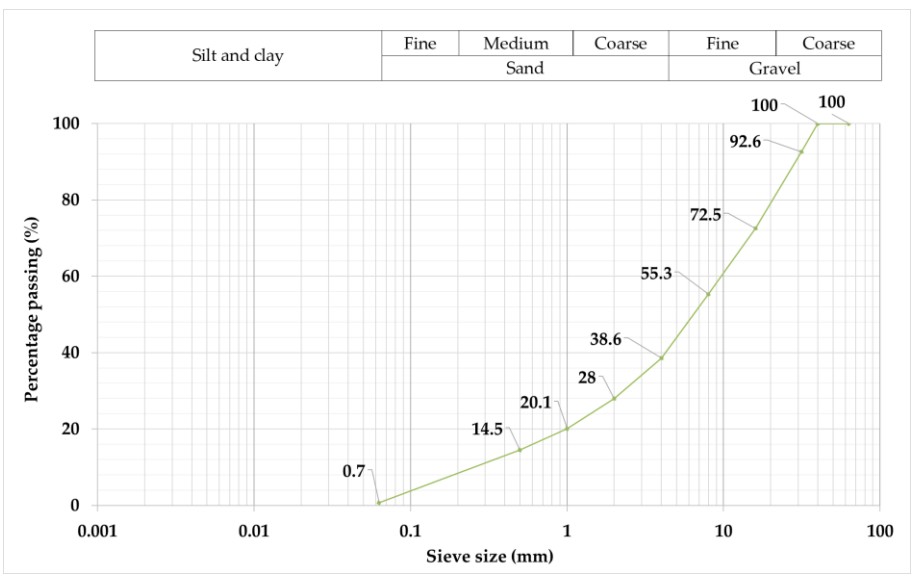

**Figure 5.** Grain size distribution of soil sample.

According to the results of Table 2, the tested soil is considered to be well graded since the coefficients $C_u$ and $C_c$ are within the specified limits. Figure 5 also includes some information about the gradation of the soil. In particular, the first part of the grain size curve is horizontal and linear, which means that there is no material with an average diameter D ≥ 40 mm. This type of material is called coarse gravel and cobbles. The next part of the curve has a steep slope, which means that the soil sample under study consists of a significant percentage of coarser material, e.g., fine and coarse gravel, while the last part has a smooth slope. Finally, the soil sample is mainly composed of sand and fine to medium gravel, while silt and clay account for up to 0.70% of the total mass.

### 3.1.2. Mechanical Properties

According to EN 13286-47 [38], the force–penetration curve was plotted for each test. Each curve follows a fifth degree polyonymic equation that fits the test results better with a coefficient of determination $R^2 \geq 0.99$. Figure 6 shows the force–penetration curves for all sample tests. All curves have a similar shape and are concave upwards, which means that there is no need for correction of the force–penetration curve.

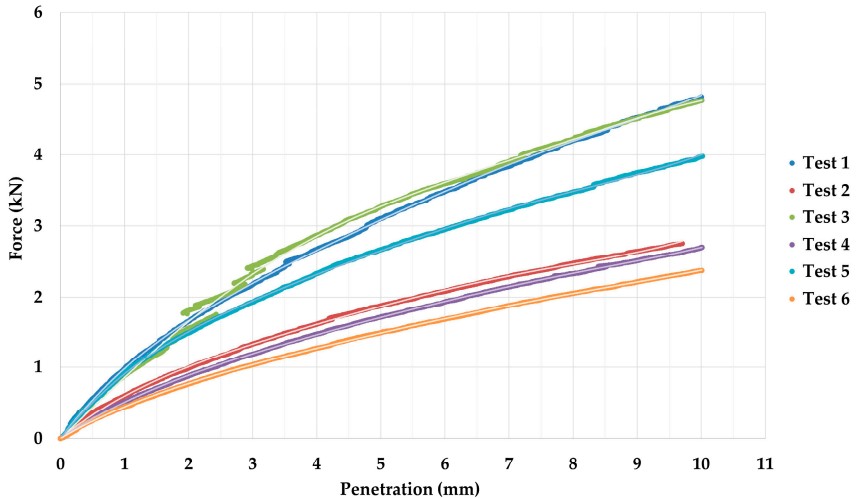

**Figure 6.** Force–penetration curve for each test.

Then, the CBR value was estimated for both 2.5 mm and 5 mm according to Equation (1), where 13.20 kN and 20 kN are the standard loads, respectively. The highest of these values was the final CBR value for each test. Table 3 summarizes the forces in kN for each test and the CBR values for each test corresponding to the required penetration.

**Table 3.** Applied forces (kN) and estimated CBR (%) values per test and standard penetration.

| No. Test | Standard Penetration (mm) | Applied Forces (kN) | CBR (%) |
|:---:|:---:|:---:|:---:|
| 1 | 2.50 | 1.95 | 14.77 |
|   | 5.00 | 3.10 | 15.50 |
| 2 | 2.50 | 1.18 | 8.94 |
|   | 5.00 | 1.87 | 9.35 |
| 3 | 2.50 | 2.03 | 15.38 |
|   | 5.00 | 3.28 | 16.40 |
| 4 | 2.50 | 1.05 | 7.95 |
|   | 5.00 | 1.73 | 8.65 |
| 5 | 2.50 | 1.75 | 13.26 |
|   | 5.00 | 2.65 | 13.25 |
| 6 | 2.50 | 0.92 | 6.97 |
|   | 5.00 | 1.49 | 7.45 |

Finally, the CBR value of the soil sample was estimated as the average of the highest values for each test and was up to 12%, with a standard deviation equal to 3.46%. Taking the average as the representative value of a data set is a common practice when the standard deviation is small or rather small, as is the case in this study. On the other hand, if one wishes to design the pavement on the safe side, a lower CBR value should be considered in the analysis. However, such considerations are beyond the scope of this study, which aims to compare the results of different relationships for MR predictions, essentially using the CBR value as the default value and MR as the variable for pavement design.

### 3.2. Determination of Resilient Modulus $M_R$

As mentioned earlier, the resilient modulus $M_R$ is used as a parameter for the bearing capacity of soil in pavement design. $M_R$ is often estimated using indirect methods, such as the estimation relationships based on the CBR measured in the laboratory, since laboratory methods, namely the triaxial repeated tests, are time-consuming and costly. Table 4 summarizes the relationships between $M_R$ (MPa) and CBR (%) that were reviewed and used in the present study.

**Table 4.** Reviewed $M_R$ (MPa)–CBR (%) correlations.

| Reference | Correlation |
|---|---|
| Heukelom and Foster [29] | $M_R = 10.8 \times CBR$ |
| Heukelom and Klomp [30] | $M_R = 10 \times CBR$ |
| Green and Hall [31] | $M_R = 37.3 \times CBR^{0.71}$ |
| Paterson et al. [32] | $M_R = 20.7 \times CBR^{0.65}$ |
| Powell et al. [33] | $M_R = 17.6 \times CBR^{0.64}$ |
| Hopkins et al. [34] | $M_R = 17.9 \times CBR^{0.874}$ |

The correlations in Table 4 were used to calculate the values of $M_R$, and Table 5 ranks the correlations associated with the magnitude of these values in descending order. The CBR value used was equal to 12%, since it was obtained as an average via the laboratory procedure.

**Table 5.** Obtained $M_R$ values in descending order (CBR = 12%).

| Correlation | $M_R$ (MPa) |
|---|---|
| Green and Hall [31] | 255 |
| Hopkins et al. [34] | 190 |
| Heukelom and Foster [29] | 162 |
| Heukelom and Klomp [30] | 150 |
| Paterson et al. [32] | 120 |
| Powell et al. [33] | 100 |

As Table 5 shows, the estimated values of $M_R$ are highly dependent on the correlation used. The variation in the values is considerable but is explained when the background for the development of these correlations is taken into account. Nevertheless, the observed differences in the values of $M_R$ definitely have an impact on the design result for a pavement.

### 3.3. Sensitivity Analysis

Accordingly, for each $M_R$-CBR correlation, the thickness $h_2$ of the subgrade overlying layer was determined. The modulus of elasticity $E_2$ used varied from 250 to 600 MPa per 50 MPa. Moreover, the coefficient k was based on Equation (12), which means that its value was between 1.5 and 7.5 to avoid the development of tensile stresses. The final value of $E_2$ was selected after testing every reasonable value so that the limits for coefficient k were not exceeded. The results for the thickness $h_2$ are given in cm, as it is implemented in practice. Table 6 show the values for coefficient k obtained from Equation (12).

**Table 6.** Values of coefficient k, based on $E_2$ values, per $M_R$–CBR correlation.

| Correlation | | Green and Hall [31] | Hopkins et al. [34] | Heukelom and Foster [29] | Heukelom and Klomp [30] | Paterson et al. [32] | Powell et al. [33] |
|---|---|---|---|---|---|---|---|
| $M_R$ (MPa) | | 255 | 190 | 162 | 150 | 120 | 100 |
| | | | | K | | | |
| | 400 | 1.57 | 2.10 | 2.47 | 2.67 | 3.32 | 4.02 |
| | 450 | 1.76 | 2.36 | 2.78 | 3 | 3.74 | 4.52 |
| $E_2$ (MPa) | 500 | 1.96 | 2.62 | 3.09 | 3.33 | 4.15 | 5.02 |
| | 550 | 2.16 | 2.88 | 3.40 | 3.67 | 4.57 | 5.52 |
| | 600 | 2.35 | 3.14 | 3.70 | 4 | 4.98 | 6.02 |

According to Table 6, the coefficient k increases in two cases: as the modulus of elasticity $E_2$ increases and as the resilient modulus $M_R$ decreases, per the correlation $M_R$–CBR. The minimum values for the whole range of $E_2$ values vary from 1.57 to 2.35, which represent the correlation of Green and Hall [31] and the highest value of $M_R$. Similarly, the maximum values for the whole range of $E_2$ values vary from 4.02 to 6.02, which represent the correlation of Powell et al. [33] and the lowest value of $M_R$. Considering a constant value of $E_2$ for each correlation $M_R$–CBR, 400 MPa requires the lowest values of k, varying from 1.57 to 4.02, while 600 MPa requires the highest values of k, varying from 2.35 to 6.02. The variation in coefficient k regarding the modulus of elasticity $E_2$ is depicted in Figure 7.

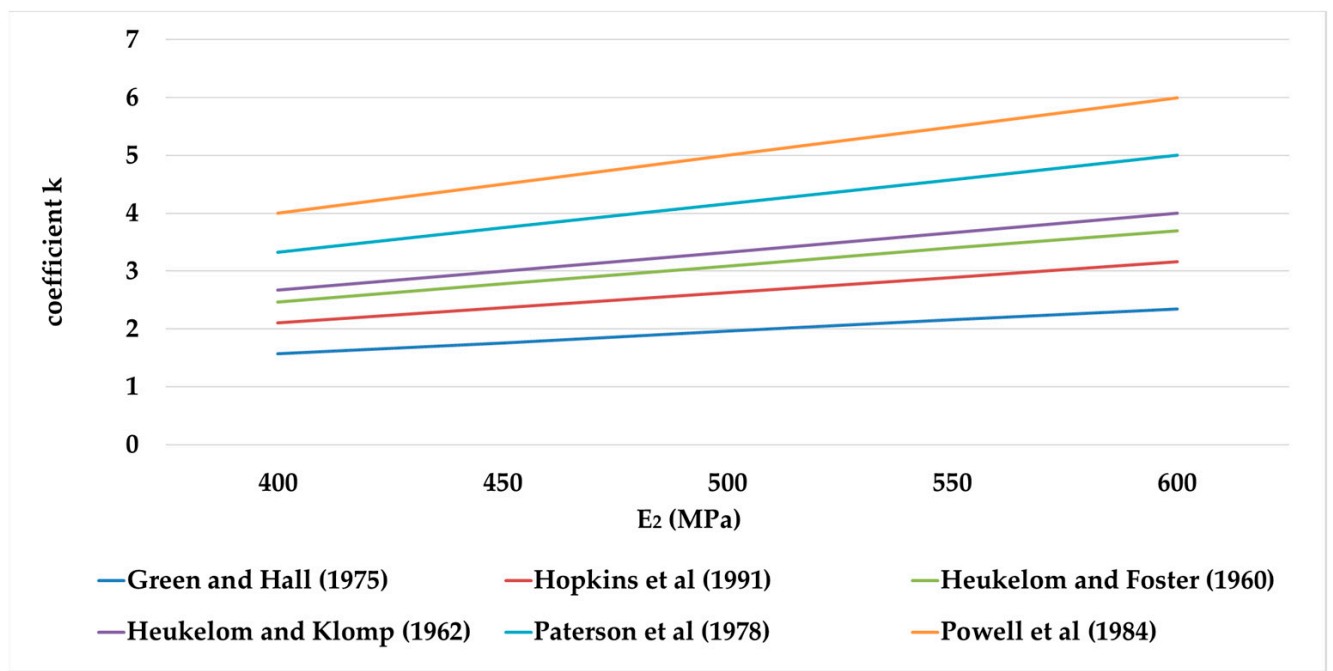

**Figure 7.** Comparison of k from the reviewed correlations.

In Figure 8, the values of $h_2$ in cm, estimated through Equation (11), are presented. The thickness $h_2$ generally increases as the modulus of elasticity increases. Considering the minimum values of k for the Green and Hall [31] correlation, $h_2$ varies between 10 and 25 cm. Overall, the Green and Hall [31] and Hopkins et al. [34] correlations require the lowest value of $h_2$ up to 25 and 50 cm, respectively, for $E_2$ to be equal to 600 MPa. The other correlations mentioned, namely Heukelom and Foster [29], Heukelom and Klomp [30], Paterson et al. [32], and Powel et al. [33], require higher values for $h_2$, ranging from 30 to 195 cm. On the other hand, assuming a constant value of $E_2$ for all correlations of $M_R$-CBR, $h_2$ increases when using a correlation from those mentioned, in order of occurrence. Finally,

the highest and lowest values of $h_2$ are obtained when the modulus of elasticity is 600 and 400 MPa, respectively.

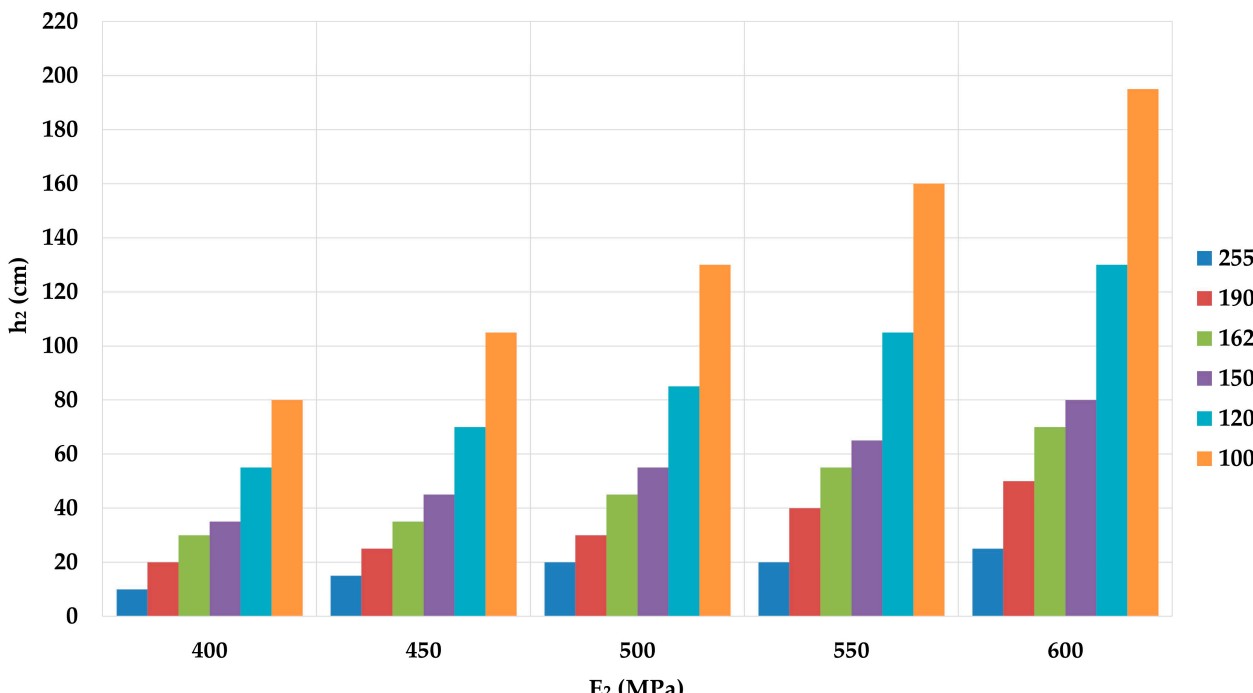

**Figure 8.** Values of thickness $h_2$ (cm), based on $E_2$ values, per $M_R$–CBR correlation.

When the Green and Hall [31] correlation was used to determine $M_R$, the smallest required thicknesses $h_2$ were obtained for both variable and constant values of $E_2$. In ad-dition, the unbound material had to have a high bearing capacity, as the modulus of elasticity $E_2$ was estimated to be 600 MPa to obtain an adequate thickness for the combined base and sub-base layer, which was 25 cm, while the rest values of $E_2$ corresponded to relatively low values of $h_2$ for a combined layer of base and sub-base (i.e., $h_2 = 10$ cm, $E_2 = 400$ MPa). Moreover, regarding the limitations of the correlation, it was applicable for CBR values up to 7% and fine-grained soils, which is indicative of a poor material. Thus, it seems rational that when a material with better bearing capacity (CBR value > 7%) is involved, the estimated value of $h_2$ may be different from the actual required value for the tested soil material applied in the field.

In addition, according to the Hopkins et al. [34] correlation, $h_2$ has a wide range of mechanically acceptable values to be applied in the field, namely 25 to 40 cm, which refers to a medium-high range of $E_2$ values of 450–550 MPa. Considering also the limitations of this correlation [34], the values of $h_2$ can be assumed to be more conservative compared to the correlation of Green and Hall [31], which has similar limitations to the correlaion of Hopkins [34]. The Heukelom and Foster [29] and Heukelom and Klomp [30] correlations provide almost similar results, as for the estimation of $h_2$, for similar ranges of $E_2$ values. As for the correlation of Heukelom and Foster [29], it provides a range of $E_2$ val-ues between 400 and 500 MPa as $h_2$ varies between 30 and 45 cm. The last results seem to be acceptable in field applications considering an economic construction as well as the limitations of the correlation (CBR values ranging from 2 to 200% based on CBR testing of different soil types). When the correlation of Heukelom and Klomp [30] is used, the overall thickness $h_2$, corresponding to $E_2$ values up to 450 MPa, has values up to 45 cm, which is considered to be an appropriate range of the $h_2$ values applied for unbound granular layers. However, if traffic volume is considered, a $h_2$ value of 55 cm is also adequate for low-volume roads. The limitations of this equation are referred to a soaked CBR value, which simulates the worst environmental conditions in the field. Therefore, since the experiment of this study refers

to the determination of the un-soaked CBR value, it can be considered that the estimated values of $h_2$ are assumed to be an over-prediction for the tested soil material.

Additionally, if the correlation of Paterson et al. [32] is used, an acceptable value for $h_2$ is 55 cm, which is the case for an $E_2$ value of 400 MPa. The rest of the values are extremely high, which is considered to be an uneconomic option for pavement design. This correlation is applicable for CBR values up to 20% and, thus, these results are con-sidered to be acceptable and can be applied in the field, especially in the case of a low-traffic road. If the correlation of Powell et al. [33] is used, the thicknesses for the whole range of $E_2$ values are extremely high and vary between 1 and 2 m. In this case, despite the correlation being applicable for CBR values between 2% and 12%, the value of $h_2$ is unreasonable, if economic construction with minimum compaction effort is sought.

## 4. Conclusions

This study addressed the parameters of bearing capacity, in particular the resilient modulus ($M_R$) and the California bearing ratio (CBR), as well as the effects of the $M_R$-CBR correlation used on the indirect determination of the bearing capacity of the sub-grade materials, which consequently determines the thicknesses of the upper layers of the flexible pavement. On this basis, the main $M_R$-CBR correlations were reviewed from the literature, while an experimental process was carried out in the laboratory on a soil material sample. By combining the $M_R$-CBR correlations and the results of the laboratory tests, a theoretical design for a flexible pavement was established focusing on the base/sub-base layer of unbound material, and a sensitivity analysis was performed in relation to the above correlations. This resulted in the following conclusions:

- In general, the thickness of the unbound granular layer ($h_2$) increases when the modulus of elasticity ($E_2$) increases, assuming a constant value of the resilient modulus ($M_R$). In addition, $h_2$ also increases when $M_R$ decreases at a constant value of $E_2$. The observed discrepancies between the results are probably due to the inadequacy of the $M_R$-CBR correlation for the soil material studied. For example, the values of $M_R$ for the Paterson et al. [32] and Powell et al. [33] correlations are of the same order of magnitude, but there are variations in the $h_2$ values due to the suitability of the first correlation for a comparatively higher quality material;
- From all correlations, it can be deduced that as $M_R$ decreases, lower values of $E_2$ are required to ensure lower values of $h_2$, taking into account a combination of economical construction and minimum compaction effort;
- According to the correlation of Green and Hall [31], the requirement of high-quality unbound material for the construction layer (base/sub-base) makes construction uneconomical despite the low thickness $h_2$ required, while for lower values of $E_2$, $h_2$ is extremely low for a total layer thickness;
- According to the correlations of Paterson et al. [32] and Powell et al. [33], an un-bound granular layer must have a comparatively large thickness $h_2$ in any case if the required value of $E_2$ is to be achieved. This is also an uneconomical solution for pavement design;
- The correlations of Heukelom and Foster [29] and Heukelom and Klomp [30] appear to be similarly applicable to the soil sample studied, since similar thicknesses of the design layer occur for the same values of $E_2$. However, due to the different limitations of each correlation, the values of $h_2$ may produce a conservative or overpredicted solution, respectively;
- As for the Hopkins et al. [34] correlation, $h_2$ has a wide range of acceptable values, namely 25 to 40 cm, which refers to a medium-high range of $E_2$ values of 450–550 MPa;
- Considering the traffic volume of flexible pavement, the correlations of Heukelom and Klomp [30] and Paterson et al. [32] may be appropriate for a low-traffic-volume road, as the total thickness of the unbound layers could be 50 to 60 cm for relatively low values of $E_2$;

- Based on the current findings, the Hopkins et al. [34] correlation assumes an appropriate overall thickness of the base and sub-base layers. In this context, low values of $h_2$ are required for correspondingly low values of $E_2$. This is considered to be the optimum combination for an economical structure, since a lower compaction effort may be required incidentally. However, considering also the limitations of each correlation, the correlation of Heukelom and Foster [29] seems to be an acceptable solution too.

In any case, it is clear that the qualitative formation of the subsoil layer is particularly important. Different types of soil material need specific design requirements, while the selection of the appropriate $M_R$-CBR correlation is crucial to ensure that the optimum solution is adopted in the design of a pavement. In this study, the limitation of the pavement design method was consequently related to the type of material used for the subgrade layer. In any case, a $M_R$-CBR correlation is, regardless, a useful tool for estimating the resilient modulus of soil materials when measuring $M_R$ is not feasible. On this basis, future research should address the need to develop a global correlation that applies to most types of soil materials, if possible.

**Author Contributions:** Conceptualization, C.P.; methodology, C.P. and M.T.; formal analysis, M.T.; investigation, C.P. and M.T.; writing—original draft preparation, M.T.; writing—review and editing, C.P. All authors have read and agreed to the published version of the manuscript.

**Funding:** This research received no external funding.

**Institutional Review Board Statement:** Not applicable.

**Informed Consent Statement:** Not applicable.

**Data Availability Statement:** Not applicable.

**Conflicts of Interest:** The authors declare no conflict of interest.

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
