# Peer review of "A Critical Comparison of Correlations for Rapid Estimation of Subgrade Stiffness in Pavement Design and Construction"

_constrmater, doi:10.3390/constrmater3010009_

Round 1

Reviewer 1 Report

A major revision is needed.

Comments:

(1) What pavement design is the study based? Different countries and districts have different types of pavement structures.

(2) How to make sure that your lab test-CBR % is similar with the field actual subgrade soil CBR%?

(3) How to calculate the Mr and CBR% based on your test?

(4) Please show your test standards in your references.

(5) Please show some penetration test figures.

(6) For the CBR% calculation, the authors can study the paper "Field investigation and numerical analysis of an inverted pavement system in Tennessee, USA". Also the pavement construction process and the compaction control can be studied from paper "Evaluating the performance of inverted pavement structure using the accelerated pavement test (APT)".

(7) What is the limitation of your design method?

(8) Please polish the language.

Author Response

The authors would like to thank the reviewer for his/her valuable comments. Detailed answers are provided below. All revisions are highlighted in the text in yellow.

A major revision is needed.

Comments:

(1) What pavement design is the study based? Different countries and districts have different types of pavement structures.

Reply: ΟΚ. Please see lines 246-248, 253-259, 263-265 and 612-613 (reference 40).

(2) How to make sure that your lab test-CBR % is similar with the field actual subgrade soil CBR%?

Reply: ΟΚ. Please see lines 107-109.

(3) How to calculate the Mr and CBR% based on your test?

Reply: ΟΚ. Please see lines 114-116, 118, 124-128, 129 and 131-132.

(4) Please show your test standards in your references.

Reply: ΟΚ. Please see lines 603-610 (references 35-38).

(5) Please show some penetration test figures.

Reply: ΟΚ. Please see lines 223-224 and new Figure 2.

(6) For the CBR% calculation, the authors can study the paper "Field investigation and numerical analysis of an inverted pavement system in Tennessee, USA". Also the pavement construction process and the compaction control can be studied from paper "Evaluating the performance of inverted pavement structure using the accelerated pavement test (APT)".

Reply: ΟΚ. Please see lines 107-109 and references 23, 24.

(7) What is the limitation of your design method?

Reply: ΟΚ. Please see lines 253-259 and 516-518.

(8) Please polish the language.

Reply: ΟΚ. It’ s done.

Reviewer 2 Report

The paper with the title "A critical comparison of correlations for rapid estimation of subgrade stiffness in pavement design and construction" is lovely. The authors showed very well this literature review in chapter one (Introduction), and after that, they used one of the materials to show what it looks like.

The second chapter is ok, but I saw some things that needed to be corrected on page 6, paragraph 209. You wrote, "d30 is the grain diameter at which 10% of mixture particles are finer." You must fix that and write 30%. In this second chapter, you wrote and showed many equations for calculating the module of elastic or module E2 and the coefficient equation.

The results and discussion were good chapters. You used one material to check grain size and showed that in figure 4. Afterwards, you went to the subsequent testing, CBR or California Bearing Ratio, where you presented results in table 3. For that table 3, I have some questions for the authors. You said the average CBR value for each test is 12%. Why you used the average value? Would you use a 15% percentile value because the values are lower? After that, you can start your calculation Mr with correlation with CBR. I saw your work on pavement construction, and when you work analyzing data for FWD, you use 15% percentile values for soil materials every time. The reason is to be on the side of safety because your 15% percentile value is smaller than the average value. Table 5 needs to be clarified because when you look at table 4, you put references and equations from literature, but why don't you put the same sequence?

In the end, the number of conclusions could be better. You wrote chapter 5, but isn't it chapter 4? I read the conclusion, and it is a very nice conclusion for this paper and this investigation.

Author Response

The authors would like to thank the reviewer for his/her valuable comments. Detailed answers are provided below. All revisions are highlighted in the text in yellow.

The paper with the title "A critical comparison of correlations for rapid estimation of subgrade stiffness in pavement design and construction" is lovely. The authors showed very well this literature review in chapter one (Introduction), and after that, they used one of the materials to show what it looks like.

The second chapter is ok, but I saw some things that needed to be corrected on page 6, paragraph 209. You wrote, "d30 is the grain diameter at which 10% of mixture particles are finer." You must fix that and write 30%.

Reply: OK. It is corrected. Please see lines 207-208.

In this second chapter, you wrote and showed many equations for calculating the module of elastic or module E2 and the coefficient equation.

The results and discussion were good chapters. You used one material to check grain size and showed that in figure 4. Afterwards, you went to the subsequent testing, CBR or California Bearing Ratio, where you presented results in table 3. For that table 3, I have some questions for the authors. You said the average CBR value for each test is 12%. Why you used the average value? Would you use a 15% percentile value because the values are lower? After that, you can start your calculation Mr with correlation with CBR. I saw your work on pavement construction, and when you work analyzing data for FWD, you use 15% percentile values for soil materials every time. The reason is to be on the side of safety because your 15% percentile value is smaller than the average value.

Reply: OK. Please see lines 362-368.

Table 5 needs to be clarified because when you look at table 4, you put references and equations from literature, but why don't you put the same sequence?

Reply: OK. Please see lines 377-378.

In the end, the number of conclusions could be better. You wrote chapter 5, but isn't it chapter 4?

Reply: OK. It’s done. Please see line 462.

I read the conclusion, and it is a very nice conclusion for this paper and this investigation.

Reply: Thank you.

Round 2

Reviewer 1 Report

Thanks for your improvement. This paper can be accepted.